# Chemical Composition of Cuticular Waxes and Pigments and Morphology of Leaves of *Quercus suber* Trees of Different Provenance

**DOI:** 10.3390/plants9091165

**Published:** 2020-09-09

**Authors:** Rita Simões, Ana Rodrigues, Suzana Ferreira-Dias, Isabel Miranda, Helena Pereira

**Affiliations:** 1Centro de Estudos Florestais (CEF), Instituto Superior de Agronomia, Universidade de Lisboa, Tapada da Ajuda, 1349-017 Lisboa, Portugal; simoes.silva2012@hotmail.com (R.S.); anadr@isa.ulisboa.pt (A.R.); hpereira@isa.ulisboa.pt (H.P.); 2Linking Landscape, Environment, Agriculture and Food (LEAF), Instituto Superior de Agronomia, Universidade de Lisboa, Tapada da Ajuda, 1349-017 Lisboa, Portugal; suzanafdias@isa.ulisboa.pt

**Keywords:** coak oak, chlorophyll, terpenes, lupeol, cuticular permeance

## Abstract

The chemical composition of cuticular waxes and pigments and the morphological features of cork oak (*Quercus suber*) leaves were determined for six samples with seeds of different geographical origins covering the natural distribution of the species. The leaves of all samples exhibited a hard texture and oval shape with a dark green colour on the hairless adaxial surface, while the abaxial surface was lighter, with numerous stomata and densely covered with trichomes in the form of stellate multicellular hairs. The results suggest an adaptive role of leaf features among samples of different provenance and the potential role of such variability in dealing with varying temperatures and rainfall regimes through local adaptation and phenotypic plasticity, as was seen in the trial site, since no significant differences in leaf traits among the various specimens were found, for example, specific leaf area 55.6–67.8 cm^2^/g, leaf size 4.6–6.8 cm^2^ and photosynthetic pigment (total chlorophyll, 31.8–40.4 µg/cm^2^). The leaves showed a substantial cuticular wax layer (154.3–235.1 µg/cm^2^) composed predominantly of triterpenes and aliphatic compounds (61–72% and 17–23% of the identified compounds, respectively) that contributed to forming a nearly impermeable membrane that helps the plant cope with drought conditions. These characteristics are related to the species and did not differ among trees of different seed origin. The major identified compound was lupeol, indicating that cork oak leaves may be considered as a potential source of this bioactive compound.

## 1. Introduction

The extracellular surface of plant leaves is covered by a hydrophobic layer known as the cuticle, which is composed primarily of cutin, an insoluble polyester of hydroxyfatty acids, glycerol and complex mixtures of waxes that are deposited within and above the structural cutin matrix [1,2,3]. Cuticular waxes vary between plant species and are composed of different organic solvent-soluble lipids, consisting of very long chain fatty acids and their derivatives including aldehydes, primary alcohols and alkanes; in some species they also contain significant amounts of pentacyclic triterpenoids [4,5,6]. They show a high degree of crystallinity, low chemical reactivity and hydrophobicity [7,8]. The composition of cuticular waxes is influenced by plant genotype, leaf side and age. Waxes differ in their functions and responses to biotic and abiotic environments.

Cuticular waxes confer properties upon the leaf surface such as nonstomatal water loss and gas exchange control, as well as protection from the external environment. Wax deposition is often a response to water stress and therefore stress resistant plants adapted to arid conditions often have thicker wax layers than those from more temperate locations or those that are susceptible to stress [9,10,11,12,13,14]. Content is not the only factor determining the properties and function of the waxy layer; knowledge of its composition is also important. However, the relationship among cuticular wax content and composition, leaf morphology and the response to different environmental conditions remains unclear.

Cork oak (*Quercus suber* L.) is an evergreen sclerophyllous tree species distributed in the western Mediterranean Basin, with a natural range including Algeria, France, Italy, Morocco, Portugal, Spain and Tunisia. Cork oak forests play an important ecological role in terms of carbon sequestration, soil protection, hydrological cycle regulation and ecosystem sustainability, while they are also of critical economic importance due to their production of cork that feeds a dedicated industrial chain [15]. However, climate change scenarios involving enhanced water deficits in the Mediterranean region threatens the ecosystem, even if cork oak shows considerable adaptability to environmental conditions and its genetic variability allows it to cope with climatic variation [16]. 

An approach to simulating a shift of climatic characteristics is provenance analysis, which enables an assessment of the phenotypic response of various populations and the identification of populations growing well and resistant to adverse environmental factors through the use of indicators or estimators of plant responses to environmental factors, that is, morphological and physiological characteristics [17]. Since the natural distribution of *Q. suber* encompasses significant environmental and geographic gradients, it can be expected that long-term natural selection and genetic drift have resulted in a high level of genetic variation among populations concerning adaptive traits such as leaf morphological area and the accumulation of cuticular wax on leaf surfaces. 

However, little information is available for *Q. suber* regarding natural variation in leaf morphology, cuticular features and phytochemical data, despite the fact that it has been demonstrated that the species possesses several mechanisms for drought tolerance including small and thick leaves, sclerophyllicity and deep tap-rooting [18,19,20,21]. Only one work was found in the literature regarding the cuticular waxes of *Q. suber* leaves [22]; for a sampling of two trees in spring and summer, the report described a composition including mainly n-alkyl esters (25–45% of the wax extract) and alkanols (18–50%), as well as alkanes, alkanals and alkanoic acids. The paper also noted significant variability in the content of cuticular waxes.

The aim of this study is to develop further knowledge of the features of *Q. suber* leaves and their variability, mainly in terms of cuticular wax quantity and composition, by using samples with different seed geographical origins that represent the area of the species’ natural distribution. Sampling was performed in the Cork Oak Provenance and Progeny Trial established in Portugal by the European Network for the Evaluation of Genetic Resources of Cork Oak for Appropriate Use in Breeding and Gene Conservation Strategies [23,24,25]. Leaves from trees with six different geographical origins (from Portugal, Spain, Italy, France, Morocco and Tunisia) were collected and the morphological parameters, pigments and cuticular wax characteristics were analysed, looking for variations which may be related to genetic or edaphoclimatic issues. This will be the first detailed study of the leaf features and cuticular wax composition of a broad sample of cork oak of different provenance, thereby providing insights into the species’ variability. 

## 2. Results 

The leaves from all of the *Quercus suber* samples showed a hard texture and oval shape with a dark green colour on the adaxial surface and a lack of trichomes, while the abaxial surface was lighter with numerous stomata and was densely covered with epidermal hairs (trichomas) in the form of stellate multicellular hairs (Figure 1). Figure 2 shows an exemplary *Q. suber* leaf cross-section observed using optical microscopy, showing the thick cuticular membrane covering the adaxial leaf side deposited on the epidermal cell layer with between-cell indentations. The trichomes and stomata are clearly visible on the abaxial side.

The average leaf features of the six cork oak provenances are presented in Table 1. Leaf size (LS) differed significantly among provenances (*p* = 0.018), ranging between 4.6 cm^2^ in the French sample (small leaves) and 6.8 cm^2^ in the Italian sample (larger leaves), with a mean coefficient of variation of leaf area of 0.39 (between 0.23 and 0.42). The mean specific leaf area (SLA) was 64.1 cm^2^/g, ranging between 55.6 cm^2^/g (Spanish sample) and 67.8 cm^2^/g (Italian sample), with a coefficient of variation in all provenances between 0.15 and 0.9 and lacking statistically significant differences (*p* = 0.243). 

The total chlorophyll content expressed on a unit per leaf area basis or per unit dry weight ranged between 40.4 µg/cm^2^ (2.53 mg/g) (Moroccan provenance) and 31.8 µg/cm^2^ (2.06 mg/g) (French provenance) with similar ratios of chlorophyll *a* to chlorophyll *b* between provenances (1.9 to 2.0). The total carotenoids concentration ranged from 8.7 µg/cm^2^ (Moroccan provenance) to 7.2 µg/cm^2^ (French provenance). Data on content of total chlorophyll and carotenoids within provenances expressed by mg/g and µg/cm^2^ differed significantly (*p* < 0.001). The average values of total chlorophylls and carotenoids in leaves of Portuguese, Moroccan and Tunisian provenance were higher than the average values of other provenances.

The average extracted yield of cuticular wax was 189.4 µg/cm^2^, with some variation among provenances, although differences were not statistically significant (*p* = 0.449), ranging from the lowest value of 136.4 µg/cm^2^ (Tunisia) to the highest value of 231.5 µg/cm^2^ (France), with coefficients of variation between 0.27 and 0.43. On a dry leaf mass basis, the cuticular waxes represented on average 24.6 mg/g, ranging between 20.7 mg/g (Spain) and 29.7 mg/g (France). Based on the driving force, the cuticular transpiration rate of the adaxial leaf surface ranged from 2.7 × 10^−4^ g/m^2^s to 5.5 × 10^−4^ g/m^2^s and the cuticular water permeance ranged from 1.6 × 10^−5^ m/s to 2.5 × 10^−5^ m/s. 

The results obtained for the cuticular wax composition of the *Q. suber* leaves are summarized in Table 2 regarding chemical families and detailed for the six provenances of *Q. suber* in Table 3. On average, the cuticular wax extracts were composed of terpenes (60.1 % of peak area), fatty acids (12.7%) and alkanes (6.1%), with a small proportion of alkanols (1.1%) and aromatics compounds (1.9%). The average proportion of identified compounds in the chromatograms was 91.2%. 

A principal component analysis (PCA) and cluster analysis (CA) were performed on the seven chemical families of epicuticular wax compounds (alkanols, alkanes, fatty acids, aromatic compounds, sterols, terpenes and others) found in 12 *Q. suber* leave samples obtained from six provenances. From the analysis of the eigenvalues of the principal components and of the scree plot, the initial hyperspace defined by seven dimensions (one for each group of compounds) was reduced to a plane defined by the first two principal components. This plane explained 56.5% of the information contained in the original data. Figure 3 shows the projections of both compound groups and samples on this plane. The first dimension (Factor 1) is positively correlated with *n*-alkanols and *n*-alkanes, while terpenes are correlated with the negative side of this axis. This indicates that samples ES-1 and Pt-2 show the highest contents of *n*-alkanols and *n*-alkanes, while samples FR-2 and MA-1 show the highest amounts of terpenes and lowest amounts of *n*-alkanols and *n*-alkanes. The second principal component (Factor 2) is highly correlated with aromatic compounds, which increases along this axis. Therefore, samples ES-2 and TU-1 show the highest content of aromatic compounds. From the analysis of Figure 3, it is difficult to identify groups of samples. Since Figure 3 shows the projections of samples on a plane and not in their original position in the hyperspace, a cluster analysis was performed on the same data to identify groups of samples (Figure 4). Again, no clear group separation was observed by CA. This confirms that there is a similar pattern for the chemical profile of the cuticular waxes of all provenances regarding chemical families.

The chemical compounds identified in the cuticular waxes of leaves from the six *Q. suber* provenances are presented in Table 3 in proportion of the total chromatogram area and are grouped by chemical family.

Triperpenes represented between 54.7% (Portuguese provenance) and 64.3% (Moroccan provenance) of the total peak area. Lupeol was the major triterpene found in all the provenances (33–40% of the compounds). Other abundant triterpenes were friedooleanan-3-ol (6.8% on average), friedelin (4.1%) and β-amyrin (5.7%), followed by smaller amounts of betulin and betulinic acid. Oleanolic and ursolic acids were also identified but in small amounts.

Sterols were present in amounts varying between 3.2% and 6.6% in the Spanish and Maroccan provenances, respectively. β-Sitosterol was the major compound from this family in all cases, while tocopherols (α, β and γ tocopherol) were present in smaller amounts.

The aliphatic compounds constituted an abundant group including fatty acids (9.2–15.2% of the compounds), n-alkanes (5.6–7.5%) and primary alcohols (0.4–2.0%), which together accounted for 17–23% of the total compounds. Among the fatty acids, C30, C28 and C16 acids were predominant, representing 36.8%, 22.7% and 12.9% of the total fatty acids, respectively. The homologous series of saturated acids with an even number of C was present from C8 to C30. Unsaturated acids only represented on average 6.9% of the total fatty acids. The chain length of the n-alkanes varied between C15 and C31, with n-octacosane (C28) and hentriacontane (C31) as the major compounds. Alkanols comprised only 0.4–1.3% of all compounds, with tretracosanol (C_24_OH) and octacosanol (C_28_OH) as the major compounds in the wax extracts. Glycerol was present in minor amounts, as well as one monoacylglycerol. Aromatic compounds comprised 1.4–2.6% of all compounds, with kaempferol, pinoresinol and hexadecy-(E)-p-coumarate as the major compounds.

## 3. Discussion

The study was conducted in a *Q. suber* provenance trial where 35 cork oak provenances covering the natural distribution range are represented, which is of utmost importance to assess the effect of seed geographic origin on adaptive traits [26,27]. In the present study, six provenances were selected that cover the broad geographic area of cork oak distribution (Portugal, Spain, France and Italy in southern Europe and Morocco and Tunisia in northern Africa).

The cork oak leaves in the trees of all the provenances showed a clear schlerophytic character (Figure 1 and Figure 2) and their morphological features (Table 1) were within the range of values reported for the species. Leaf sizes (4.6–6.8 cm^2^) were similar to those reported by Mediavilla et al. [28] for *Q. suber* leaves taken from different orientations in the canopy (5.5–7.4 cm^2^) and the 7.1 cm^2^ reported by Prats et al. [20]. The specific leaf area values (55.6–67.8 cm^2^/g) were of the same order of magnitude as those obtained in adult leaves of *Q. suber* growing under contrasting environments and located in different positions and orientations of the canopy (50.0 to 126.0 cm^2^/g) [28,29,30,31,32]. 

When comparing the six cork oak provenances, two sets could be defined in relation to leaf area (Table 1), with ES, FR and TU provenances characterized by smaller leaves and PT, IT and MA provenances characterized by larger leaves. Specific leaf area (SLA) for the six provenances showed a low coefficient of variation (CV 0.14), suggesting that they developed a similar degree of leaf sclerophylly. This finding is in line with most of the few available studies on *Q. suber*. Rzigui et al. [31] observed that there was no difference in SLA values between two provenances in Tunisia (Gafour and Feija) with contrasting environments (125 and 126 cm^2^/g) and Daoudi et al. [33] reported that SLA did not differ significantly between three humid, semi-arid and sub-humid provenances in Algeria, although SLA was lower under conditions of water deficiency. Lobo-do-Vale et al. [32] reported an adaptive response to drought with an SLA reduction (85 cm^2^/g in a mild year and 65 cm^2^/g in a dry year), while Aranda et al. [34] observed that, while irrigation had no significant effect, SLA decreased with increasing irradiance, thereby improving the potential for carbon uptake relative to transpiration water loss. SLA plays an important role in linking plant carbon and water cycles and is sensitive to environmental change [35,36], with leaf traits being influenced by a combination of species, climate and soil factors [37,38,39].

In the present study, the observed amounts of the total chlorophyll, expressed per unit of leaf area or per unit of dry weight (Table 1), were consistent with those reported in the available studies for *Q. suber*, for example, the results of Ramírez-Valiente et al. [30] for leaves of open-pollinated trees from populations in Morocco, Portugal and Spain (2.68 mg/g, 2.64 mg/g and 2.74 mg/g respectively) and 55.7 µg/cm^2^ and 56.0 µg/cm^2^ total chlorophyll in sun and shade leaves of 40-year-old *Q. suber* trees [40]. Mediavilla et al. [28] also observed that the chlorophyll content differed among canopy orientations, with lower values on the west side (80 vs. 110 µg/cm^2^), probably due to a direct effect of excess radiation at the time of the day with highest leaf temperatures and lowest water potential on the west facing leaves. 

A substantial average wax layer of 189.4 µg/cm^2^ (or 24.6 mg/g) covered the leaves of the six *Q. suber* provenances (Table 1) without any provenance effect. The values are slightly above the 125 µg/cm^2^ reported by Martins et al. [22] for the cuticular wax of young leaves of *Q. suber*. A higher amount of cuticular wax is observed in *Q. suber* leaves than in other *Quercus* species. For instance, young leaves of *Q. ilex* (holm oak, a persistent leaf oak) have 71 µg/cm^2^ of cuticular wax [22], *Q. robur* (pedunculate oak, deciduous tree growing in cooler climates with less hydric stress) has 59 µg/cm^2^ [41] and *Q. petraea* (sessile oak) has 101.5–134.5 µg/ cm^2^ [42]. Similar values were also reported for *Q. polymorpha* (Mexican white oak) leaves, with a wax layer of 199.4 µg/cm^2^ [43].

The high wax content of the cork oak leaves suggests that cuticular characteristics may be associated with adaptation to the local environmental conditions of high temperatures and water deficit. Accumulation of cuticular wax is considered an important strategy against drought in many plant species, providing an essential barrier to protect plants from drought stress [44,45]. In this study, the cork oak did not differ in their responses to environmental conditions by provenance, indicating that the accumulation of cuticular waxes on the leaves, albeit depending on the species, mainly responds to the specific environmental conditions.

Despite the role of the cuticle as a water barrier, there is still a movement of water through the cuticle between the outer cell wall of the epidermis and the surrounding atmosphere, giving the cuticle some permeability [46]. This movement is based on a simple diffusion process along a gradient of the water’s chemical potential with the water molecules being absorbed at one interface, following a random path through the cuticle in a mainly lipophilic chemical environment and desorbed at the other interface [47]. The degree to which cuticles transferred water was measured in the adaxial leaf surface of the *Q. suber* samples, with an average cuticular permeance of 1.8 × 10^−5^ m/s (Table 2). No information is available on the cuticular permeance of *Q. suber* leaves but comparison with other *Quercus* species shows that this value is quite low, in agreement with its adaptation to hot and dry environments. The available values for cuticular permeances range from 3.6 × 10^−5^ m/s for *Q. ilex*, 7.3 × 10^−5^ m/s for *Q. rubra,* 10 × 10^−5^ m/s for *Q. coccifera*, to 27 × 10^−5^ m/s for *Q. sessiliflora* [47]. The permeances determined with isolated cuticular membranes were from 3.8 × 10^−5^ m/s to 7.4 × 10^−5^ m/s for *Q. petraea* [48].

Our results on the chemical composition of the dichloromethane extract of *Q. suber* leaves (Table 2 and Table 3) show that the majority of compounds are pentacyclic triterpenoids (mainly lupeol) and that long-chain aliphatic components are mainly fatty acids (mainly in C30, C28 and C16). The extraction method may have a determining role in the extent of compound solubilization, namely, on the amount of terpenes in relation to aliphatic compounds. Since the objective in the present study was to extract the total wax layer, including epicuticular and intracuticular waxes from both sides of the leaf, an intensive 6 h extraction with dichloromethane was conducted. 

The only work to our knowledge regarding the composition of cuticular waxes in young *Q. suber* leaves [22] applied a quick surface extraction method by dipping and shaking the leaves for a few seconds with chloroform and this explains the compositional differences to our results. In fact, this extract contained only small amounts of triterpenoids (triterpenone, friedelin), although the aliphatic composition was similar to that found in the present study: 4–27% *n*-alkanes; 18–50% even chain amphiphilic compounds (*n*-alkan-1-ols); up to 25% *n*-alkanals; <5% *n*-alkanoic acids; and 25–45% *n*-alkyl esters [22]. It was reported that triterpenoids are located almost exclusively in the intracuticular wax compartment and therefore require a more intensive extraction procedure [49,50].

The cuticular wax composition appears to be related to the species, since no statistical significant differences were found between cork oak provenances and there was no clustering of provenances by chemical families, based on PCA and cluster analysis (Figure 3 and Figure 4).

*Q. suber* leaf wax contains the same lipid classes as *Q. ilex* and *Q. robur* but with different distribution patterns. In a *Q. ilex* leaf, the most abundant wax components are C22 and C24 *n*-alkanoic acids (38%) and *n*-alkan-1-ols (43–54%), with small amounts of the triterpenols α- and β-amyrin [30]. In a *Q. robur* leaf wax, the dominating classes are alcohols (about 70% of the wax with chain lengths ranging from C16 to C34, with tetracosanol as a main component), fatty acids (20% of the wax, especially with chain length of C14 and C22), aldehydes (28%, of the wax with chain lengths from C20 to C32 with C26 and C28 as the main components) and several triperpenoids (8% of the wax, taraxerol, β-amyrin, α-amyrin and lupeol were identified) [41,51]. 

A comparison with *Fagus* and *Castanea* species, also members of the Fagaceae family to which *Q. suber* belongs, shows that the epicuticular leaf wax of *Castanea sativa* consists of a homologous series of wax lipids (wax esters, aldehydes, primary alcohols and fatty acids) and large amounts of triterpenoids (α- and β-amyrin and lupeol) [52], while that of *Fagus sylvatica* contains only wax lipids, without any triterpenoids [53].

The cuticular wax of *Q. suber* leaves is rich in triterpenoids that can be obtained as an extract after solubilization. Over the last three decades, extensive research has revealed important pharmacological applications of triterpenoids with potential uses in new functional foods, drugs, cosmetics and healthcare products. Lupeol was studied for the treatment of various diseases, including skin wounds and various medicinal properties of lupeol have been reported, including anti-inflammatory, antioxidant, anti-diabetic and anti-mutagenic effects [54,55,56]. 

*Quercus suber* leaves are a promising and highly available source of triterpenes, since large quantities of leaves are generated each year from silvicultural practices (e.g., pruning), which can be used to produce chemicals using environmentally friendly extraction processes [57]. Our results estimate a potential extraction yield per kg of dry leaves of *Q. suber* of 15 g triterpenes, of which 9 g is lupeol.

## 4. Material and Methods

### 4.1. Sampling

The study was carried out on a provenance trial of *Quercus suber* L. at Herdade do Monte Fava, Santiago do Cacém, near Setúbal, in central Portugal (38° 00′ N, 08°07′ W, altitude 79 m). The site has a Mediterranean climate, with hot and dry summers (total year rainfall, 556.6 mm; summer accumulated rainfall, 19.4 mm; annual average temperature, 15.8 °C; average minimum temperature of the coldest month, 4.3 °C; average maximum temperature of the hottest month, 31.3 °C) and the soil has a sandy texture [26]. The trial was established in March 1998, as part of an international network on cork oak genetic resources funded by an EU Concerted Action Fair 202 [23]. The plant material included in this field trial resulted from a seed collection conducted during the autumn of 1996 from 35 cork oak populations that covered the species’ natural range; seedlings were raised from these seeds with a common protocol in one nursery, planted in the trial field and the trees were allowed to grow until leaves were sampled [23,26].

The sampling carried out for the present study included 21-year-old trees from six provenances: Portugal, Spain, Italy, France, Morocco and Tunisia. Detailed information on the location and climate data from the original seed collection sites of the studied provenances is given in Table 4 as well as that of the trees’ growing site. Leaves were randomly sampled from two trees of each provenance in March 2019. The leaves were collected from different branches on the southern exposed side of the crown, in the lower part of the canopy up to a height of approximately 2 m, making up a total sample per tree of about 100 leaves.

### 4.2. Morphological Variables

The morphological variables were measured in 20 leaves that were randomly selected from leaves sampled for each of the two trees of each provenance. Leaves were digitalized and analysed using WinSEEDLE^TM^ 2011. The leaves were oven-dried at 70 °C to a constant mass and the total dry mass per leaf was determined. Specific leaf area (SLA, cm^2^/g) was calculated as the ratio between the measured leaf area and dry weight; this was used as an indirect index of sclerophylly. The sclerophylly index was calculated as (IE) = leaf dry mass (g)/2 × leaf area (dm^2^) and defines sclerophylly as IE > 0.6 and mesophylly as IE < 0.6, according to Rizzini [58].

### 4.3. Determination of Leaf Pigment Contents

Chlorophyll *a* and *b* were determined in six disks of known area taken from leaves of each of the two trees per provenance. The tissue samples were homogenized in the dark during 24 to 36 h, with 3 mL of dimethylformamide solution. Chlorophyll *a* and *b*, carotenoids and total chlorophyll were quantified in the supernatant phase by using a spectrophotometer and readings were taken at 663.8, 646.8 and 480.0 nm, respectively, according to the methodology described by Lichthenthaler [59] and using the equations and specific absorption in the wavelength reported by Wellburn [60]. The leaf pigments chlorophyll a and b, total chlorophyll and carotenoids were expressed as µg/cm^2^ of leaf area.

### 4.4. Cuticle Permeability Assay 

The cuticular permeance was determined by the measurement of water loss through the adaxial, astomatous leaf surface. Leaf samples were rehydrated by submerging the petioles of the leaves in water at room temperature for 12 h. The leaf petioles and the abaxial leaf surface were sealed with paraffin wax to ensure that water transpiration only occurred via the stomata-free adaxial leaf surface. Leaf weight was measured immediately afterwards. Leaf samples were then placed in boxes over silica gel and placed in an incubator with control the surrounding temperature (25 °C) and weighed repeatedly over a 4 h period. The transpiration rate (flux of water vapor; *J*, in g m^−2^ s^−1^) was obtained from the change in fresh weight of the samples (Δ*W*, in g) over time (Δ*t*, in s) and surface area (*A*, in m^2^): $$J=\frac{∆W}{∆t\;\times A}.$$

The leaf surface area (*A* leaf) of the adaxial surface was obtained by scanning the leaf surface.

The cuticular permeance *P* (in m s^−1^) and minimum leaf conductance *g_min_* can be obtained from the transpiration rate *J* and the driving force *Δc* of water across the cuticle according to
$$P=\;g_{min}=\frac{J}{c_{WV}\left(a_{leaf}-a_{air}\right)}=\frac{J}{c_{WV}\;∆c}.$$

The driving force (Δc) is the difference between the concentration of water vapour in the leaf interior and the surrounding atmosphere (g m^–3^), resulting in permeability parameters in m s^–1^. Since during cuticular transpiration the diffusion of water occurs in the solid phase of the cuticle, atmospheric pressure has no influence and, consequently, using the concentration-based driving force is appropriate. The resulting permeability parameters were described in m s^–1^. The water activity of leaf samples (*a_leaf_*) was assumed to be unity [49]. The humidity of the air was controlled by silica gel, resulting in a water activity (*a_air_*) close to zero. Therefore, the driving force for water loss through transpiration was identical to the density of water vapour at saturation in the air (c*_WV_* at 25 °C was 23.07 g m^−3^; [61]) [14,62]. 

### 4.5. Extraction of Cuticular Waxes 

Cuticular wax was extracted from whole fresh leaves with dichloromethane in a Soxhlet apparatus over 6 h using a sample of 20 leaves per tree. After extraction, the solvent was evaporated. This extraction method yields a total wax mixture containing both epicuticular and intracuticular waxes that cover both sides of the leaf. 

The amount of soluble cuticular lipids was determined from the mass difference of the extracted leaves after drying at 105 °C and was expressed on a leaf surface area and dry weight basis (the ratio between wax in μg and the two-sided leaf surface area in cm^2^, obtained by digitalization).

### 4.6. Cuticular Wax Composition 

The cuticular wax (obtained as dichloromethane extracts from 20 leaves per tree) was analyzed using gas-chromatography mass spectrometry (GC-MS). Two mg of each leaf extract was taken and derivatized in 120 μL of pyridine; the compounds with hydroxyl and carboxyl groups were trimethylsilylated into trimethylsilyl (TMS) ethers and esters, respectively, by adding 80 μL of bis(trimethylsily)-trifluoroacetamide (BSTFA). The reaction mixture was heated at 60 °C for 30 min in an oven. The derivatized extracts (1 μL) were immediately analyzed by GC-MS (EMIS, Agilent 5973 MSD, Palo Alto, CA, USA), with an ionization energy of 70 eV and the MS source was kept at 220 °C under the following GC conditions: Zebron 7HGG015-02 column (30 m, 0.25 mm; ID, 0.1 μm film thickness), with injector at 280 °C. The column temperature was initially held at 50 °C for 1 min, raised to 150 °C at a rate of 10 °C min^−1^, then to 300 °C at 4 °C min^−1^, to 370 °C at 5 °C min^−1^ and at 8 °C min^−1^ until it reached 380 °C; this was followed by an isothermal period of 5 min. Compounds were identified as TMS derivatives by matching their mass spectra with a GC-MS spectral library (Wiley, NIST) and by comparing their fragmentation profiles with published data [63,64]. Two replicates were made per extract. 

### 4.7. Structural and Anatomical Observations

For scanning electron microscopy (SEM), small pieces of fresh- and air-dried leaves were fixed to sample holders and observed with a scanning electron microscope (TM 3030 Plus Hitachi).

The cuticular membrane was observed in a thin leaf cross section by optical microscopy. The leaves were impregnated with DP 1500 polyethylene glycol and cross sections of approximately 10 µm thickness were cut with a rotary microtome (Medite M530). The sections were stained with safranin and astral blue and were mounted in Euparal. Observations were made using a light microscope (Leica DM LA) and the photomicrographs were taken with a Nikon Microphot-FXA.

### 4.8. Statistical Analyses

To compare leaf functional traits among provenances, one-way analysis of variance (ANOVA) was performed. Duncan’s post-hoc tests were used to analyse pairwise differences between provenances. Statistical significance was set at *p* < 0.05. All statistical analyses were performed using the Sigmaplot^®^ (Version 11.0, Systat Software, Inc., Chicago, IL, USA).

Principal component analysis (PCA) and cluster analysis (CA) were performed to analyse the chemical composition of cuticular wax from leaves of *Quercus suber* trees of different provenance. Using PCA, the hyperspace was defined by the original variables and dimension reduction was carried out using the significant principal components (new axes). These new axes were correlated with the original variables. Thus, the samples were plotted onto a reduced space where similar samples could be grouped. Agglomerative hierarchical CA, using the Euclidean distance and single-linkage method, was carried out to assess the existence of groups of samples suggested by PCA [65]. PCA and CA were performed with the Statistica^TM^ software, version 6, from Statsoft (Tulsa, OK, USA).

## 5. Conclusions 

The results obtained for *Quercus suber* leaves from six different provenances suggest the adaptive value of leaf features under a Mediterranean climate, namely, specific leaf area, leaf size and photosynthetic pigment, highlighting their potential role in dealing with varying temperatures and rainfall regimes through local adaptation and phenotypic plasticity. Cork oak leaves possess a substantial cuticular wax layer that forms a nearly impermeable membrane which is a tool to cope with adverse drought related with environmental conditions. The composition of the cuticular wax also favours the hydrophobicity of the layer by inclusion of very long-chain alkanes and alkanoic acids (e.g., C28 and C30). These characteristics are species specific and did not differ across provenances.

Triterpenes are the major component of the cuticular wax complex and contain a high proportion of lupeol. Thus, the leaves of cork oak represent a potential source for this interesting bioactive compound. The chemical composition of the cuticular wax did not differ among provenances, which may be ascribed to similar water stress conditions in the Mediterranean region, which probably has a stronger effect than genetic variability in *Q. suber*.

## Figures and Tables

**Figure 1 plants-09-01165-f001:**
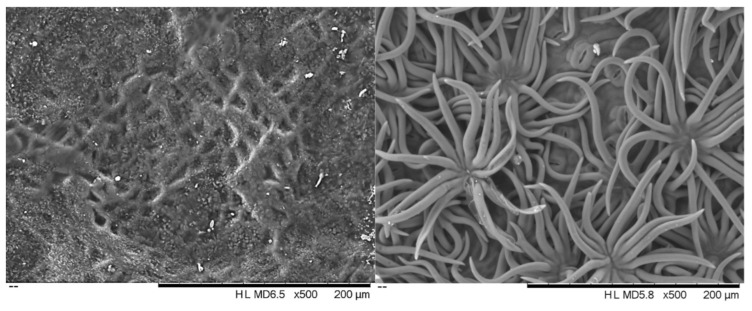
Scanning electron micrographs of the adaxial (**left**) and abaxial (**right**) surface of *Quercus suber* leaves.

**Figure 2 plants-09-01165-f002:**
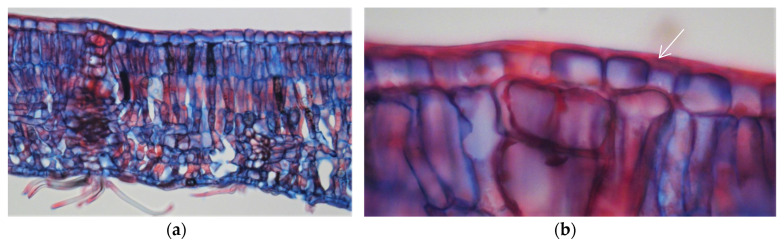
Optical microscopy photographs of a cross-section of a *Quercus suber* leaf (**a**) and an enlargement of the adaxial side showing the cuticular membrane (**b**). Arrows indicate the cuticular structures covering the epidermal cell layer.

**Figure 3 plants-09-01165-f003:**
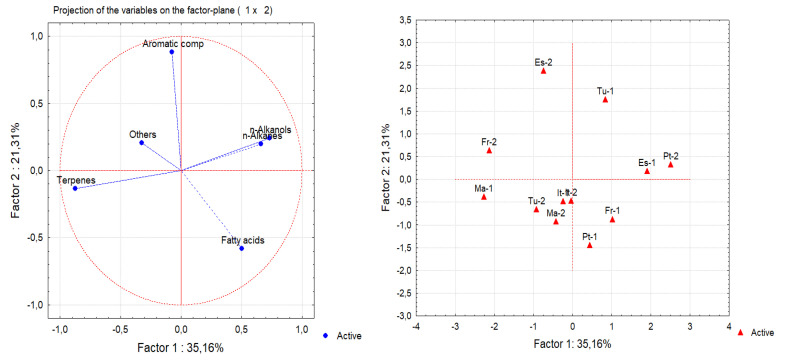
Principal component analysis (PCA) of the chemical classes of epicuticular wax compounds from leaves of 12 samples from six provenances: (left) score plots of the original variables; (right) sample plot (ES—Spain; Fr—France; IT—Italy; MA—Morocco; PT—Portugal; TU—Tunisia).

**Figure 4 plants-09-01165-f004:**
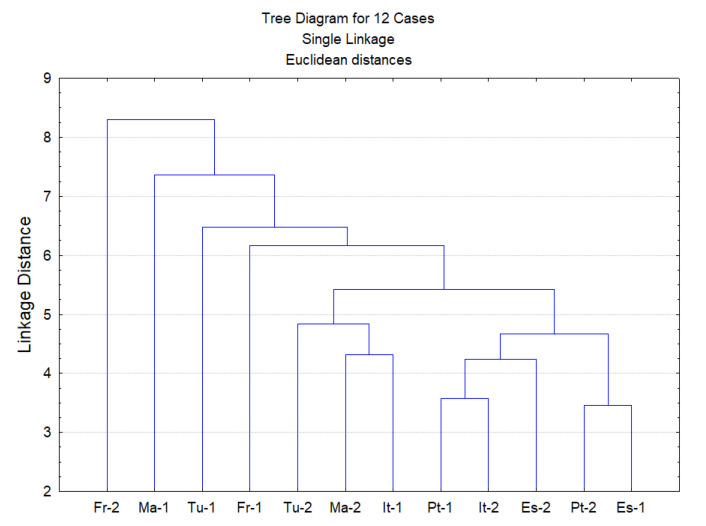
Dendrogram obtained by cluster analysis of the family classes of epicuticular wax compounds from 12 leave samples from six provenances (ES—Spain; FR—France; IT—Italy; MA—Morocco; PT—Portugal; TU—Tunisia).

**Table 1 plants-09-01165-t001:** Morphological and physiological characterization of leaves of cork oak (*Quercus suber*) from six provenances. The provenances are Portugal (PT35), Spain (ES11), Italy (IT3), France (FR3), Morocco (MA27) and Tunisia (TU32). Mean and standard deviation of 20 leaves per provenance.

Leaf Features/Provenance Code	PT35	ES11	IT3	FR3	MA27	TU32
Leaf size (LS; cm^2^)	6.2 ± 2.1a	5.1 ± 2.1b	6.8 ± 2.7a	4.6 ± 1.9b	6.4± 2.8a	5.6 ± 2.1b
Specific leaf area (SLA; cm^2^/g)	61.5 ± 18.5a	55.6 ± 17.6a	67.8 ± 22.7a	63.1 ± 19.4 a	64.7 ± 27.9a	63.7 ± 28.9a
Sclerophylly index (g/dm^2^)	0.9 ± 0.3a	1.0 ± 0.4a	0.8 ± 0.3a	0.9 ± 0.4a	0.9 ± 0.4a	0.9 ± 0.4a
Cuticular wax content (µg/cm^2^)	212.5 ± 40.3a	182.2 ± 60.2a	201.0 ± 60.4a	231.5 ± 64.4a	173.0 ± 80.6a	136.4 ± 45.9b
Cuticular wax content (mg/g)	25.9 ± 0.5a	20.7 ± 0.5a	25.7 ± 0.6a	29.7 ± 0.7a	24.2 ± 0.4a	21.6 ± 0.4a
**Photosynthetic Pigments**						
Chlorophyll a (µg/cm^2^)	26.6 ± 1.6a	23.9 ± 3.2b	24.1 ± 2.0b	21.1 ± 1.4b	26.4 ± 2.9a	24.8 ± 0.7b
Chlorophyll b (µg/cm^2^)	13.3 ± 2.1a	12.0 ± 1.9a	12.1 ± 1.4a	10.7 ± 1.4b	14.0 ± 1.8a	12.7 ± 0.5a
Chlorophyll a/b ratio	2.0 ± 0.3a	2.0 ± 0.1a	2.0 ± 0.1a	2.0 ± 0.1a	1.9 ± 0.1a	2.0 ± 0.1a
Total chlorophyll (µg/cm^2^)	40.0 ± 3.4a	35.9 ± 5.0b	36.3 ± 3.4b	31.8 ± 1.8b	40.4 ± 4.5 a	37.5 ± 1.0a
Total chlorophyll (mg/g)	2.5 ± 0.2a	2.1 ± 0.3a	2.3 ±0.2a	2.1 ± 0.1a	2.5 ±0.3a	2.7 ± 0.1a
Total carotenoids (µg/cm^2^)	8.4 ± 0.4 a	7.9 ± 0.7b	7.5 ± 0.8b	7.2 ± 0.1b	8.7 ± 0.1a	8.3 ± 0.2a
Chlorophyll/carotenoids ratio	0.2 ± 0.03a	0.3 ± 0.03a	0.3 ± 0.04a	0.3 ± 0.01a	0.2 ± 0.03a	0.2 ± 0.01a
**Cuticular Water Permeability**						
T_cut_ × 10^−4^ (g/m^2^s)	5.5	2.7	4.9	3.3	4.8	4.0
P × 10^−5^ (m/s)	2.4	1.2	2.1	1.4	2.1	1.7

(T_cut_) cuticular transpiration rate at maximum driving force. (P) cuticular permeance. Means in each row followed by the same letter are not significantly different at *p* < 0.05.

**Table 2 plants-09-01165-t002:** Chemical class composition of the cuticular wax of leaves from six *Quercus suber* provenances, as a percentage of the total peak areas in the GC-MS chromatograms. Mean and standard deviation of 12 samples.

Family	Percent of Total Compounds
*n*-Alkanols	1.10 ± 0.56
*n*-Alkanes	6.07 ± 0.72
Fatty acids	12.73 ± 2.41
Aromatic compounds	1.94 ± 0.44
Sterols	5.32 ± 1.23
Terpenes	60.05 ± 3.65
Others	3.76 ± 1.98
Total identified compounds	91.15 ± 3.95

**Table 3 plants-09-01165-t003:** Chemical composition (% of all chromatogram peak areas) of the cuticular waxes of leaves from six *Quercus suber* provenances (mean of two trees per provenance). The provenances are Portugal (PT35), Spain (ES11), Italy (IT3), France (FR3), Morocco (MA27) and Tunisia (TU32).

Compound/Provenance Code	PT35	ES11	IT3	FR3	MA27	TU32	Whole Leaves
****n-Alkanols****	**2.03**	**1.33**	**0.90**	**0.93**	**0.35**	**1.05**	**1.10 ± 0.56**
hexadecan-1-ol (C_16_OH)	0.21	0.17	0.08	0.00	0.05	0.08	0.10 ± 0.08
octadecan-1-ol (C_18_OH)	0.00	0.17	0.03	0.00	0.00	0.00	0.03 ± 0.07
tretracosan-1-ol (C_24_OH)	1.43	0.75	0.53	0.66	0.15	0.60	0.69 ± 0.42
octacosan-1-ol (C_28_OH)	0.39	0.24	0.27	0.27	0.15	0.37	0.28 ± 0.09
**n-Alkanes**	**5.78**	**5.92**	**5.57**	**6.09**	**5.55**	**7.46**	**6.06 ± 0.72**
n-pentadecane (C_15_)	0.10	0.06	0.04	0.04	0.08	0.15	0.08 ± 0.04
n-Pentacosane (C_25_)	0.38	0.33	0.36	0.42	0.42	0.44	0.39 ± 0.04
n-heptacosane (C_27_)	0.20	0.28	0.28	0.21	0.10	0.27	0.22 ± 0.07
n-octacosane (C_28_)	3.65	3.73	3.07	3.96	3.56	4.38	3.73 ± 0.44
1-nonacosene (C_29(2_))	0.06	0.39	0.00	0.10	0.19	0.25	0.17 ± 0.14
n-nonacosane (C_29_)	0.47	0.10	0.33	0.49	0.29	0.40	0.35 ± 0.14
n-hentriacontane (C_31_)	0.93	1.04	1.49	0.87	0.92	1.57	1.14 ± 0.31
**Fatty Acids**	**15.22**	**11.70**	**14.19**	**14.93**	**11.12**	**9.24**	**12.73 ± 2.41**
**Saturated**	**14.57**	**11.50**	**13.50**	**13.48**	**9.93**	**8.59**	**11.93 ± 2.33**
octanoic acid (C_8:0_)	0.13	0.17	0.13	0.04	0.08	0.13	0.11 ± 0.05
nonanoic acid (C_9:0_)	0.05	0.21	0.28	0.17	0.03	0.06	0.13 ± 0.10
nonadioc acid (C_9(2)_)	0.04	0.11	0.05	0.40	0.33	1.07	0.33 ± 0.39
decanoic acid (C_10:0_)	0.11	0.09	0.22	0.00	0.05	0.09	0.09 ± 0.07
dodecanoic acid (C_12:0_)	0.13	0.16	0.21	0.12	0.10	0.12	0.14 ± 0.04
tetradecanoic acid (C_14:0_)	0.25	0.15	0.35	0.27	0.28	0.31	0.27 ± 0.07
hexadecanoic acid (C_16:0_)	1.04	1.37	1.01	1.66	1.95	1.62	1.44 ± 0.37
octadecanoic acid (C_18:0_)	0.21	0.23	0.17	0.31	0.27	0.29	0.25 ± 0.05
eicosanoic acid (C_20:0_)	0.35	0.23	0.26	0.43	0.28	0.29	0.31 ± 0.07
docosanoic acid (C_22:0_)	0.67	0.44	0.39	0.69	0.38	0.45	0.50 ± 0.14
tetracoscanoic acid (C_24:0_)	0.55	0.32	0.40	0.82	0.20	0.23	0.42 ± 0.23
hexacosanoic acid (C_26:0_)	0.88	0.53	0.71	0.87	0.64	0.27	0.65 ± 0.23
octacosanoic acid (C_28:0_)	4.03	2.51	3.19	3.39	2.38	1.22	2.79 ± 0.98
triacontanoic acid (C_30:0_)	6.13	4.98	6.13	4.31	2.96	2.44	4.49 ± 1.56
**unsaturated**	**0.65**	**0.20**	**0.69**	**1.45**	**1.19**	**0.65**	**0.81 ± 0.45**
9,12-octadecadienoic acid (C_18:2_)	0.12	0.06	0.20	0.39	0.37	0.17	0.22 ± 0.13
9,12,15-octadecatrienoic acid (C_18:3_)	0.53	0.14	0.49	1.06	0.82	0.48	0.59 ± 0.32
**Aromatic Compounds**	**1.39**	**2.57**	**1.77**	**1.77**	**1.41**	**1.74**	**1.78 ± 0.43**
benzoic acid	0.06	0.13	0.11	0.06	0.03	0.08	0.08 ± 0.04
4-hydroxybenzaldehyde	0.11	0.23	0.07	0.17	0.14	0.12	0.14 ± 0.06
vanillin	0.01	0.06	0.06	0.00	0.11	0.03	0.05 ± 0.04
4-(2-hydroxyethy) phenol	0.13	0.20	0.16	0.48	0.35	0.07	0.23 ± 0.15
methyl p-coumarate, trans	0.06	0.16	0.00	0.16	0.12	0.12	0.10 ± 0.06
quercetin/myricetin	0.19	0.05	0.27	0.05	0.21	0.08	0.14 ± 0.09
kaempferol	0.13	0.30	0.55	0.00	0.09	0.49	0.26 ± 0.22
pinoresinol	0.33	0.46	0.32	0.62	0.04	0.50	0.38 ± 0.20
2,3-dihydrobenzofuran	0.26	0.17	0.12	0.17	0.07	0.18	0.16 ± 0.06
hexadecy-(E)-p-coumarate	0.38	0.99	0.23	0.24	0.32	0.25	0.40 ± 0.29
**Sterols**	**5.11**	**3.16**	**5.14**	**6.36**	**6.62**	**5.54**	**5.32 ± 1.23**
β-tocopherol/γ-tocopherol	0.34	0.42	1.07	0.22	0.29	0.62	0.49 ± 0.31
α-tocopherol	0.58	0.22	0.41	0.66	0.69	0.16	0.45 ± 0.23
β-sitosterol	4.19	2.52	3.66	5.48	5.64	4.76	4.38 ± 1.18
**Terpenes**	**55.64**	**57.53**	**62.56**	**58.24**	**64.61**	**60.70**	**59.88 ± 3.36**
**diterpenes**	**1.00**	**0.29**	**1.21**	**1.59**	**1.30**	**0.55**	**0.99 ± 0.49**
phytol	0.46	0.21	1.00	1.18	0.86	0.38	0.68 ± 0.39
α-tocopherolquinone	0.54	0.08	0.21	0.41	0.44	0.17	0.31 ± 0.18
**pentacyclic triterpenes**	**54.64**	**57.24**	**61.35**	**56.65**	**64.31**	**60.15**	**59.06 ± 3.54**
α-amyrin	1.01	0.92	0.97	1.00	1.09	0.73	0.95 ± 0.12
β-amyrin	5.14	4.82	4.63	5.68	8.26	5.30	5.64 ± 1.34
lupeol	37.76	34.37	38.59	32.91	37.22	40.49	36.89 ± 2.79
friedooleanan-3-ol	3.95	7.59	8.25	7.29	7.85	5.82	6.79 ± 1.62
friedelin	2.94	5.39	3.79	4.60	4.86	3.12	4.12 ± 0.99
betulin	1.42	1.40	1.44	2.02	2.25	1.75	1.71 ± 0.36
oleanolic acid	0.43	0.63	0.61	0.92	0.42	0.76	0.63 ± 0.19
betulinic acid	1.27	1.31	1.78	1.73	1.94	1.83	1.64 ± 0.28
ursolic acid	0.72	0.81	1.29	0.50	0.42	0.35	0.68 ± 0.35
**Others**	**3.48**	**3.59**	**2.66**	**8.21**	**3.29**	**3.48**	**4.12 ± 2.03**
myo-inositol/ Scyllo-inositol	1.55	1.30	1.60	3.46	1.06	1.01	1.66 ± 0.91
D (-) fructofuranose	0.39	0.33	0.00	0.79	0.39	0.38	0.38 ± 0.25
D (-) fructopyranose	0.43	0.36	0.00	0.79	0.28	0.51	0.40 ± 0.26
glycerol	0.46	0.75	0.45	0.56	0.59	0.29	0.52 ± 0.16
6,10,14 trimethtylpentadecan-2-one	0.09	0.52	0.23	1.91	0.40	0.82	0.66 ± 0.66
1-Linoleylglycerol	0.08	0.01	0.14	0.25	0.27	0.11	0.14 ± 0.10
**Total identified compounds**	**88.65**	**85.80**	**92.77**	**96.52**	**93.94**	**89.21**	**91.15 ± 3.95**

Bold is highlighted the family of compounds.

**Table 4 plants-09-01165-t004:** Characterization of the trial site and identification of the *Quercus suber* provenances with collection site location, T_m_ annual average air temperature (°C) and the long-term annual average precipitation (PPT, mm) (adapted from Varela [23]) and characterization of the provenance trial of *Quercus suber* (Herdade do Monte Fava).

Provenance Code	Country of Seed Collection	Latitude	Longitude	Altitude (m)	T_m_ (°C)	PPT (mm)
**PT35**	Portugal, Ermidas do Sado	38° 00′ N	8° 70′ W	79	15.8	557
**ES11**	Spain, Alpujarras	36° 50′ N	3° 18′ W	1300	13.0	742
**IT13**	Italy, Puglia	40° 34′ N	17° 40′ E	45	16.6	588
**FR3**	France, Landes	43° 45′ N	1° 20′ W	20	12.3	870
**MA27**	Morocco, Rif Occidental I.2	35° 07′ N	5° 16′ W	300	n.a.	1280
**TU32**	Tunisia, Mekna	36° 57′ N	8° 51′ W	12	17.9	948
**Monte da Fava**	Portugal, Santiago do Cacém	38° 00’ N	8° 07’ W	79	15.8	557

n.a.: not available.

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
