# Peer review of "Chemical Composition of Cuticular Waxes and Pigments and Morphology of Leaves of Quercus suber Trees of Different Provenance"

_plants, 2020, doi:10.3390/plants9091165_

Round 1

Reviewer 1 Report

The authors state that they are already conducting a ‘longitudinal’ analysis in different years. I encourage them to include these additional results in this manuscript to improve the scientific value, instead of fragment their findings in different papers. Furthermore, the chemical composition of waxes is done considering only 1 extract/provenance: it is definitely not representative of the provenance itself and this elaboration cannot be accepted by the scientific community. I already suggested to consider only the average values of the 6 extracts, without discriminate the proveniences. My opinion is to reject this manuscript until the results of the next years of observations will be included, allowing a reliable statistical elaboration of the results.

Author Response

Reviewer 1

The authors state that they are already conducting a ‘longitudinal’ analysis in different years. I encourage them to include these additional results in this manuscript to improve the scientific value, instead of fragment their findings in different papers. Furthermore, the chemical composition of waxes is done considering only 1 extract/provenance: it is definitely not representative of the provenance itself and this elaboration cannot be accepted by the scientific community. I already suggested to consider only the average values of the 6 extracts, without discriminate the proveniences. My opinion is to reject this manuscript until the results of the next years of observations will be included, allowing a reliable statistical elaboration of the results.

We included the average values, as requested by the reviewer in Table 2.  However, it is not true that only one extract per provenance was considered; instead two trees per provenance were sampled and analysed individually. And we complied with the other 3 reviewers in expanding the statistical analysis regarding the provenances, which we did.

We disagree with the reviewer on the postponing of publishing these results and waiting for the finalization of the longitudinal analysis. As we know this time-based evolution should consider at least 3 yearly cycles, meaning that we had to wait still a considerable time. We agree on the relevance of such a study and we had in fact already started it. When we will publish these longitudinal results we will probably couple the seasonal variation of composition of the cuticular waxes with that of the cutin, thereby allowing an integrative reasoning on the variation of the chemical composition of the cuticular membrane of the cork oak. Therefore, we are not “slicing” the results but rather constructing a knowledge base on the matter.

We also stress that very little is now known on the cuticular waxes of the cork oak, namely on a detailed chemical composition as it was made here (as well as on other leaf features). Therefore we think that making available this knowledge to the scientific community is important.

Reviewer 2 Report

I have revised the previous version of this work and I think that the paper has been improved with statistical analysis in Table 1, some more details on Fig. 2, PCA and cluster analysis. In addition, the discussion has been partially revised. I would suggest some minor changes in the discussion to include and highlight the results of the new statistical approach with PCA and cluster analysis. I suggest the authors to submit the revised version with marked changes (for ex. in red), otherwise it is very difficult to follow the changes. 

Author Response

Reviewer 2

I have revised the previous version of this work and I think that the paper has been improved with statistical analysis in Table 1, some more details on Fig. 2, PCA and cluster analysis. In addition, the discussion has been partially revised. I would suggest some minor changes in the discussion to include and highlight the results of the new statistical approach with PCA and cluster analysis. I suggest the authors to submit the revised version with marked changes (for ex. in red), otherwise it is very difficult to follow the changes. 

Done. We improved the discussion as suggested

Reviewer 3 Report

After second revision of the manuscript, I found it has really improve, answering main questions and critical points highlighted in the first revision, so I do find enough improvements to get it accepted.

Author Response

Reviewer 3

After second revision of the manuscript, I found it has really improve, answering main questions and critical points highlighted in the first revision, so I do find enough improvements to get it accepted.

Thank you for helping us to improve the manuscript

Reviewer 4 Report

A brief summary

A review of the manuscript entitled: „Chemical composition of cuticular waxes, pigments and morphology of leaves of Quercus suber trees from different provenances”is very interesting and promising. This paper is descriptive, material paper. New aspects should be emphasized. The manuscript describes the chemical composition of cuticular wax, pigments and morphological features of Quercus suber leaves in provenances.

Based on this general evaluation and the specific comments, reported below, I recommend a minor revisions of the manuscript. I have few comments and suggestions, which might improve the manuscript.

Broad comment

This is an interesting and extensive study of  stream water temperature across a large sample of streams and sub-catchments. The data sets is good and nicely presented.

The aim of study need better organization and clarity. In its present form, the aim sounds very descriptive. Also, logical consistency between aims and results-discussion sections should be improved. Minor concern - It is not clear on what basis the authors  suggest an adaptive  value of leaf features eg. specific leaf area (Abstract section and Conclusions). The mean  SLA values are not significantly different (p = 0.243). Please, clarify. 

Specific comment

Results

Table 1 - Please provide an explanation of abbreviations provenance code. These explanations are only given in the Table 4.

Material and methods

The methods are not described accurately enough. There is no information about the total number of leaves. How many leaves were used for chemical composition?

L. 328-337 How is the relationship between seed collection conducted during the autumn of 1996 and Quercus suber leaves samples, collected in March 2019? Should be described in more detail.

L. 351 – ‘projected leaf area’ or measured leaf area? Please, clarify

Author Response

Reviewer 4

A brief summary

A review of the manuscript entitled: „Chemical composition of cuticular waxes, pigments and morphology of leaves of Quercus suber trees from different provenances”is very interesting and promising. This paper is descriptive, material paper. New aspects should be emphasized. The manuscript describes the chemical composition of cuticular wax, pigments and morphological features of Quercus suber leaves in provenances.

Based on this general evaluation and the specific comments, reported below, I recommend a minor revisions of the manuscript. I have few comments and suggestions, which might improve the manuscript.

Done, as detailed below. The new aspects were emphasized at the end of the introduction. Thanks

Broad comment

This is an interesting and extensive study of  stream water temperature across a large sample of streams and sub-catchments. The data sets is good and nicely presented.

The aim of study need better organization and clarity. In its present form, the aim sounds very descriptive. Also, logical consistency between aims and results-discussion sections should be improved. Minor concern - It is not clear on what basis the authors  suggest an adaptive  value of leaf features eg. specific leaf area (Abstract section and Conclusions). The mean SLA values are not significantly different (p = 0.243). Please, clarify. 

Done. This was clarified.

Specific comment

Results

Table 1 - Please provide an explanation of abbreviations provenance code. These explanations are only given in the Table 4.

Done. This was included in the title of Table 1

Material and methods

The methods are not described accurately enough. There is no information about the total number of leaves. How many leaves were used for chemical composition?

Done. The total number of leaves collected (100 per tree) and extracted (20 leaves per tree) had already been included in the text. This was now included in the subsection of composition as well as the specification of the number of repetitions.

  1. 328-337 How is the relationship between seed collection conducted during the autumn of 1996 and Quercus suberleaves samples, collected in March 2019? Should be described in more detail.

Done. Included in the manuscript : “; seedlings were raised from these seeds, with a common protocol in one nursery, planted in the trial field and the trees allowed to grow until the present sampling of leaves”

  1. 351 – ‘projected leaf area’ or measured leaf area? Please, clarify

Done. Changed in the manuscript